# MRFF-YOLO: A Multi-Receptive Fields Fusion Network for Remote Sensing Target Detection

**Danqing Xu and Yiquan Wu \***

College of Electronic and Information Engineering, Nanjing University of Aeronautics and Astronautics, Nanjing 211106, China; xudanqing@nuaa.edu.cn

**\*** Correspondence: imagestrong@nuaa.edu.cn; Tel.: +86-137-7666-7415

**Abstract:** High-altitude remote sensing target detection has problems related to its low precision and low detection rate. In order to enhance the performance of detecting remote sensing targets, a new YOLO (You Only Look Once)-V3-based algorithm was proposed. In our improved YOLO-V3, we introduced the concept of multi-receptive fields to enhance the performance of feature extraction. Therefore, the proposed model was termed Multi-Receptive Fields Fusion YOLO (MRFF-YOLO). In addition, to address the flaws of YOLO-V3 in detecting small targets, we increased the detection layers from three to four. Moreover, in order to avoid gradient fading, the structure of improved DenseNet was chosen in the detection layers. We compared our approach (MRFF-YOLO) with YOLO-V3 and other state-of-the-art target detection algorithms on an Remote Sensing Object Detection (RSOD) dataset and a dataset of Object Detection in Aerial Images (UCS-AOD). With a series of improvements, the mAP (mean average precision) of MRFF-YOLO increased from 77.10% to 88.33% in the RSOD dataset and increased from 75.67% to 90.76% in the UCS-AOD dataset. The leaking detection rates are also greatly reduced, especially for small targets. The experimental results showed that our approach achieved better performance than traditional YOLO-V3 and other state-of-the-art models for remote sensing target detection.

**Keywords:** remote sensing target detection; multi-scale; multi-reception field; densely connected network; Res2 block; YOLO-V3

## 1. Introduction

The high-altitude remote sensing images [1–4] obtained by satellites and aircrafts are widely used in military, navigation, disaster relief, etc. So, remote sensing target detection [5–7] is becoming an important research hotspot. The interferences of the light changes, environment, and other complex backgrounds in remote sensing images make remote sensing targets hard to be detected. At present, there are still some problems such as low detection accuracy, error detections, and missed detections.

In order to realize remote sensing target detection, researchers have made unremitting efforts. The algorithm of elliptical Laplace operator filtering based on Gaussian scale space was proposed in 2010 [8]. It treated the vehicle targets as elliptical class objects and employed elliptic operators in different directions to perform convolution filtering with the targets. Then, the k-nearest-neighbor classifier was used to separate the false targets. In 2015, a new method for remote sensing target detection was proposed by Naoto Yokoya et al. [9]. It combined feature detection based on sparse representation with generalized Huff transform. Then, by adopting the method of learning the dictionary of targets and backgrounds, the sparse image representation of specific classes was constantly supplemented. Finally, the remote sensing target detection was realized. According to the detection of high-resolution optical satellite ship targets, Buck et al. [10] firstly considered the use of frequency domain characteristics to extract the candidate areas of ship targets. Then, the length–width ratio of the superstructure and the

length ratio of the whole ship were adopted to extract ship targets. The above algorithms achieved good results. However, when facing the remote sensing targets under complex background, these conventional algorithms still had some problems such as low detection accuracy, error detections, and missed detections.

Recently, with the development of computational power, applications based on deep learning [11–13] have made great achievements and have been widely used in all sorts of fields. The great success of AlexNet [14–16] in the image classification competition in 2012 made target detection based on deep learning become a new hotspot, and various state-of-the-art target detection models were proposed. Generally speaking, they can be divided into two categories: the region-based algorithms and the regression-based algorithms. The former ones are represented by Region-Based Convolutional Neural Network (R-CNN) [17], which was firstly proposed by Ross Girshick et al.; these include Fast R-CNN [18–20], Faster R-CNN [21–23], Mask R-CNN [24,25], etc. The latter ones mainly include YOLO series such as YOLO-V1 [26], YOLO-V2 [27–29], YOLO-V3 [30,31], YOLO-V3 tiny [32], etc., and Single Shot Multibox Detector (SSD) series such as SSD [33], Deconvolutional SSD (DSSD) [34], and Feature Fusion SSD (FSSD) [35,36], etc.

The region-based target detection algorithms firstly enumerate the candidate boxes on the feature maps and then classify them in a fine way to obtain the detection results. Therefore, they usually have the advantage of high accuracy. The shortcomings of them are also obvious. Since the detection process is divided into two steps, the speed is slow, the storage cost is high, and the model cannot be compressed. So, it is hard for them to meet the real-time requirements. In contrast, the regression-based target detection algorithms overcome the shortcomings of the region-based algorithms. The location information and category information of the targets are predicted by the network directly. So, they have a good real-time performance and are widely used in the engineering applications.

YOLO (You Only Look Once) was firstly proposed by Joseph Redmon and Ali Farhadi et al. in 2015. The main contributions of YOLO are are follows. (1) YOLO regards target detection as a problem of regression. (2) The structure of YOLO is very concise. It is a one-stage target detection model, and it predicts the information of location and category of bounding boxes at the same time by Convolutional Neural Network (CNN) [37–39] directly. (3) YOLO inputs the images into the network to get the final detection result directly, so YOLO has higher speed. (4) YOLO inputs the entire image into the network for detection. So, it can encode global information and reduce errors in regarding the background as targets. Due to the above advantages, YOLO is popular among researchers.

YOLO-V3 is the latest version of YOLO. As an open source target detection network, YOLO-V3 has obvious advantages in speed and accuracy, and it achieves great performance on multi-scale target detection. A large amount of target detection algorithms based on YOLO-V3 have been proposed since YOLO-V3 first appeared. Reference [40] adopted a new feature extraction network and rounded ground truth to detect the rounded targets such as a tomato, which is aimed at making better use of feature information, Reference [41] adopted a new style of connection for residual units; Reference [42] added another detecting layer to detect small targets.

However, when facing remote sensing targets with complex background, they cannot be detected efficiently due to (1) feature underutilization and (2) the loss of target receptive fields. In order to take full advantage of the feature information and detect remote sensing targets effectively, a more effective YOLO-V3-based model (MRFF-YOLO) was proposed. The new network adopted the improved 'Res2 block', 4th detection layer, and DenseNet. We compared our MRFF-YOLO with other state-of-the-art target detection algorithms on RSOD and UCS-AOD datasets to evaluate the performance of remote sensing target detection.

The main contributions in this paper are as follows. (1) In order to improve the performance of feature extraction and realize receptive fields' fusion simultaneously, the proposed 'Res2 block' was adopted to replace the deep-level residual units in the original feature extraction network of YOLO-V3. (2) With the aim of avoiding gradient fading, the convolutional layers in the detection layers are

replaced by densely connected network (DenseNet). (3) To enhance the performance of detecting the remote sensing targets with small size, the 4th scale was added to the framework of YOLO-V3.

The rest part of this paper is organized as follows. (1) In Section 2, we introduced the framework of YOLO-V3. (2) In Section 3, we detailed the improvements of our approach. (3) In Section 4, experimental verification was given to verify the effectiveness of our approach. (4) Finally, we gave the conclusion of this paper in Section 5.

## 2. Introduction to YOLO

YOLO is the most popular regression-based target detection algorithm due to its conciseness and high speed. Compared with the region-based algorithms such as Fast R-CNN and Faster R-CNN, YOLO is suitable for engineering applications due to the simple and efficient network. Since the advent of YOLO, YOLO has evolved from YOLO-V1 to YOLO-V2 and YOLO-V3.

### 2.1. The Fundamental of YOLO

When detecting the targets, YOLO will firstly divide the input image into $S \times S$ grid cells. The grid cell that the center of the target falls in will be responsible for detecting it. For each grid cell, YOLO predicts $B$ bounding boxes. For each bounding box, YOLO predicts five values: four values for the location of the bounding box, and one value for the confidence of the bounding box. The confidence can be defined as $P(Object) \times IOU_{prid}^{truth}$. Confidence measures two aspects: one is whether the target lies in the bounding box, and the other is the bounding box's accuracy in predicting the position of the target. If no target lies in the bounding box, then the confidence of the bounding box is 0. If the bounding box contains the target, then $P(Object) = 1$, and the confidence will be the IOU (Intersection-Over-Union) between the bounding box and ground truth. In addition, YOLO predicts $C$ categories for each grid cell and a set of conditional probabilities for each grid cell: $P(Class_i|Object)$.

From the above, the output of the network contains a total of $S \times S$ grid cells. Each grid cell predicts $B$ bounding boxes. Each bounding box predicts five values. In addition, each grid cell predicts $C$ categories. So, the size of the output tensor of the network is $S \times S \times (B \times 5 + C)$.

### 2.2. The Principle of YOLO-V3

To extract deeper information of the network, YOLO upgrades the feature extraction network from Darknet19 to Darknet53. YOLO makes heavy use of residual units. Instead of pooling layers, YOLO adopts convolutional layers with stride = 2 to implement down-sampling. The structure of YOLO-V3 is shown in Figure 1.

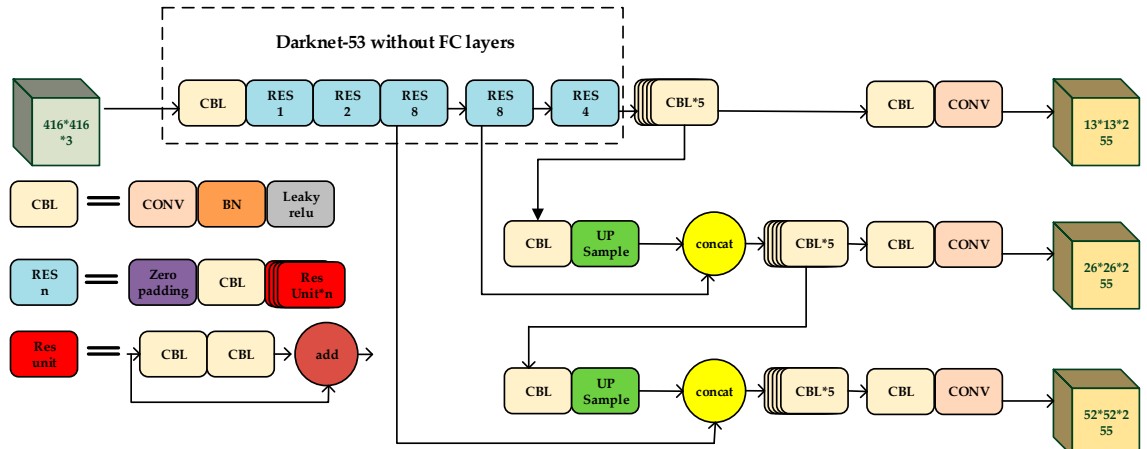

**Figure 1.** The structure of You Only Look Once (YOLO)-V3 network.

The feature extraction network down-samples the image to 32×, which signifies that the size of the output feature map is 1/32 the size of the input image. To enhance the performance of detecting small targets, the detection was carried out at the feature map down-sampled by 32×, the feature map down-sampled by 16×, and the feature map down-sampled by 8×, respectively. Up-sampling is adopted due to the reason that the deeper the network, the better the effect of feature expression. For example, in the case of detecting targets with the feature map down-sampled by 16×, if the 4th down-sampling layer is directly used for detection, the effect is generally not good. So, the network doubles the size of the feature map down-sampled by 32× by up-sampling a with step size of 2. In this way, the dimensions of the two feature maps remain the same. Then, the network concatenates the two feature maps to achieve feature fusion. Similarly, we do the same for the other detection layers.

The final outputs of YOLO-V3 are three scales: $13 \times 13$, $26 \times 26$, and $52 \times 52$, which are responsible for the detection of big targets, medium-sized targets, and small targets, respectively.

In YOLO-V3, the loss function can be divided into three parts: coordinate prediction error, IOU error, and classification error [43].

The coordinate prediction error is defined as:

$$
\begin{aligned}
Error_{coord} \quad &= \lambda_{coord} \sum_{i=1}^{s^2} \sum_{j=1}^{B} I_{ij}^{obj} \left[ (x_i - \overline{x}_i)^2 + (y_i - \overline{y}_i)^2 \right] \\
&+ \lambda_{coord} \sum_{i=1}^{s^2} \sum_{j=1}^{B} I_{ij}^{obj} \left[ (w_i - \overline{w}_i)^2 + (h_i - \overline{h}_i)^2 \right]
\end{aligned}
\tag{1}
$$

In Equation (1), $S^2$ represents the number of the grid cells of each scale. $B$ denotes the number of bounding boxes for each grid. $I_{ij}^{obj}$ represents whether there is a target that falls in the $j$-th bounding box of the $i$-th grid cell. $(\overline{x}_i, \overline{y}_i, \overline{w}_i, \overline{h}_i)$ and $(x_i, y_i, w_i, h_i)$ represent the center coordinate, height, and width of the predicted box and the ground truth, respectively.

The IOU error is defined as:

$$
\begin{aligned}
Error_{\text{IOU}} \quad &= \sum_{i=1}^{s^2} \sum_{j=1}^{B} I_{ij}^{obj} (C_i - \overline{C}_i) \\
&+ \lambda_{noobj} \sum_{i=1}^{s^2} \sum_{j=1}^{B} I_{ij}^{obj} (C_i - \overline{C}_i)
\end{aligned}
\tag{2}
$$

In Equation (2), $C_i$ and $\overline{C}_i$ denote the true and predicted confidence, respectively.

The third part is the classification error:

$$
Error_{cls} = \sum_{i=1}^{s^2} \sum_{j=1}^{B} I_{ij}^{obj} \sum_{c \in classes} (p_i(c) - \hat{p}_i(c))^2.
\tag{3}
$$

In Equation (3), $p_i(c)$ refers to the true probability of the target, while $\hat{p}_i(c)$ refers to the predicted value.

From the above, the final loss function is shown in Equation (4):

$$
\begin{aligned}
Loss \quad &= Error_{coord} + Error_{\text{IOU}} + Error_{cls} \\
&= \lambda_{coord} \sum_{i=1}^{s^2} \sum_{j=1}^{B} I_{ij}^{obj} \left[ (x_i - \overline{x}_i)^2 + (y_i - \overline{y}_i)^2 \right] \\
&+ \lambda_{coord} \sum_{i=1}^{s^2} \sum_{j=1}^{B} I_{ij}^{obj} \left[ (w_i - \overline{w}_i)^2 + (h_i - \overline{h}_i)^2 \right] \\
&+ \sum_{i=1}^{s^2} \sum_{j=1}^{B} I_{ij}^{obj} (C_i - \overline{C}_i) \\
&+ \lambda_{noobj} \sum_{i=1}^{s^2} \sum_{j=1}^{B} I_{ij}^{obj} (C_i - \overline{C}_i) \\
&+ \sum_{i=1}^{s^2} \sum_{j=1}^{B} I_{ij}^{obj} \sum_{c \in classes} (p_i(c) - \hat{p}_i(c))^2
\end{aligned}
\tag{4}
$$

## 3. Methodology

Even the state-of-the-art YOLO-V3 model still has poor performance in detecting remote sensing targets due to the complex background and small size of the targets. For real-time remote sensing

target detection, it is necessary to increase the receptive fields and extract features more effectively without deepening the network.

Therefore, based on the original YOLO-V3 model, several improvements are proposed for the feature extractor and detection layers.

### 3.1. The Feature Extractor Based on Res2Net

In order to alleviate the problem of gradient fading on the premise of deepening the network, the feature extractor of YOLO-V3 employs the structure of ResNet. The Darknet53 of YOLO-V3 contains five residual blocks. Each residual block consists of one or a set of multiple residual units. The structure of the residual unit is exhibited in Figure 2.

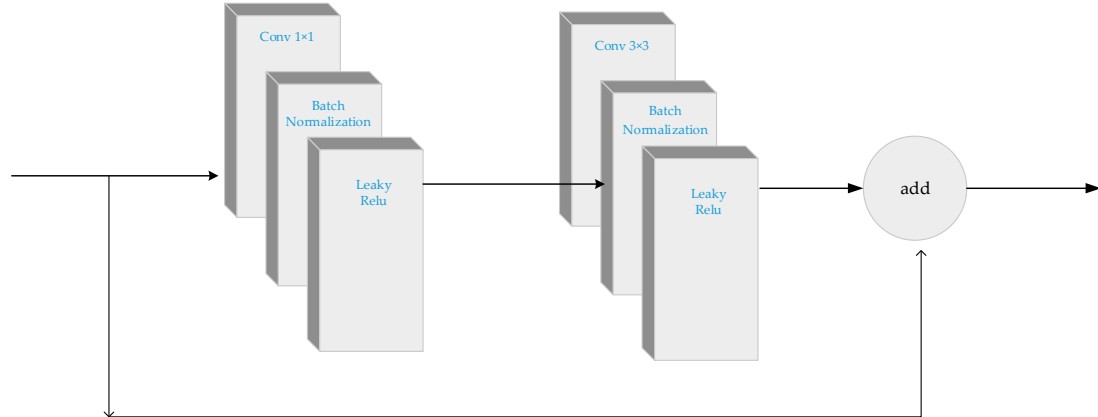

**Figure 2.** The structure of the residual unit.

The residual blocks in YOLO-V3 overcome the problem of gradient fading when deepening the feature extraction network and enhancing the performance of feature expression. Representing features on multiple scales is important for many visual tasks. However, ResNet still represents multi-scale features in a hierarchical manner, which makes the features within each layer underutilized. To solve this problem, Gao et al. [44] proposed a new connection method for the residual units to extract features. In this method, the author constructed hierarchical residual class connections in a single residual block and proposed a new building block, which is named Res2Net. Res2Net represents multi-scale features with finer granularity and increases the range of receptive fields at each layer. Borrowing the core idea of Res2Net, we added several tiny residual terms to the original residual units to increase the receptive fields of each layer. Compared with the residual unit, the structure of our proposed 'Res2 unit' is shown in Figure 3.

In the 'Res2 unit', we divide the input feature map into N sub-features (N = 4 in this paper) on average after the $1 \times 1$ convolutional layer. Each sub-feature is represented as $x_i (i = 1, 2, \ldots N)$. Each $x_i$ is in the same size, but it only contains $1/S$ number of channels compared with the input feature map [45]. $K_i()$ represents the $3 \times 3$ convolutional layer. We represent $y_i$ as the output of $K_i()$. So, $y_i$ is represented as:

$$y_i = \begin{cases} x_i & i = 1; \\ K_i(x_i) & i = 2; \\ K_i(x_i + y_{i-1}) & 2 \leq i \leq N. \end{cases} \tag{5}$$

In particular, $y_1, y_2, y_3, y_4$ can be expressed as (∗ represents convolution):

$$\begin{cases} y_1 = x_1 \\ y_2 = x_2 * (3 \times 3\text{Conv}) \\ y_3 = (x_3 + x_2 * (3 \times 3\text{Conv})) * (3 \times 3\text{Conv}) \\ y_4 = (x_4 + (x_3 + x_2 * (3 \times 3\text{Conv})) * (3 \times 3\text{Conv})) * (3 \times 3\text{Conv}) \end{cases}. \tag{6}$$

In this paper, we set $N$ as the controlling parameter, which means that the number of input channels can be divided into multiple feature channels on average. The larger $N$ is, the stronger the multi-scale capability will have for the network. In this way, we will get an output of different sizes of receptive fields.

Compared with the residual unit, the improved 'Res2 unit' makes better use of contextual information and can help the classifier detect small targets and the targets subject to environmental interference more easily. In addition, the extraction of features at multiple scales enhance the semantic representation of the network.

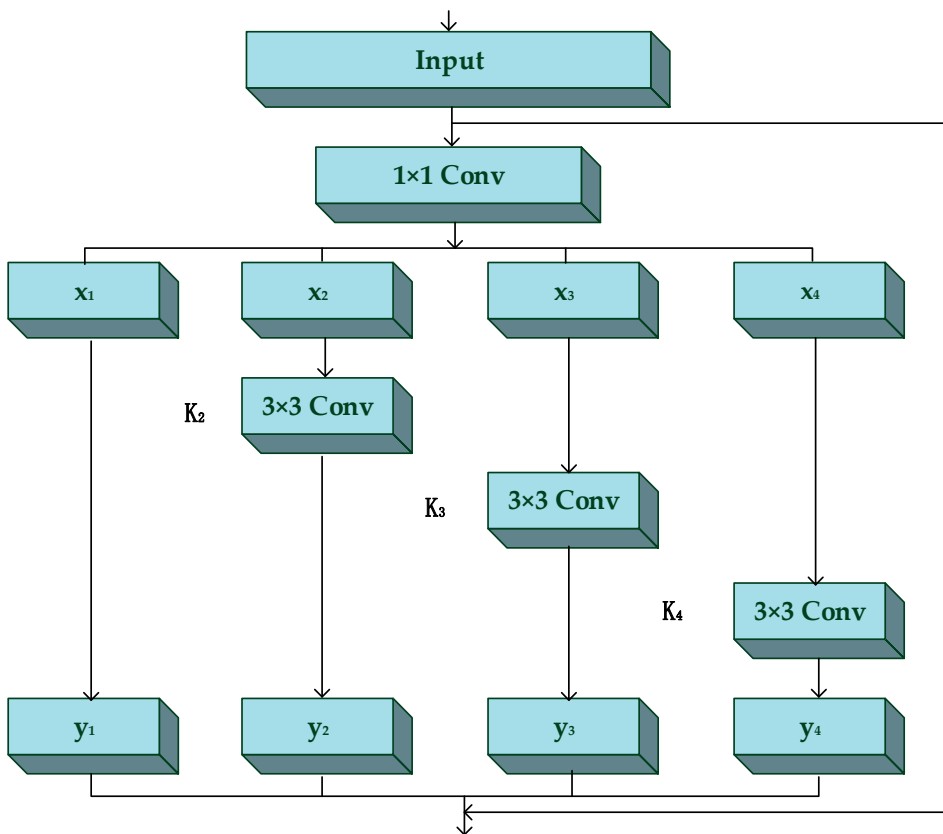

**Figure 3.** The structure of the Res2 unit.

### 3.2. Densely Connected Network for Detecting Layers

The structure of YOLO-V3 in Figure 1 shows that there are six convolutional layers in each detecting layer. In order to avoid gradient fading, we introduce the concept of densely connected networks (DenseNet).

DenseNet [46–49] was firstly proposed by Huang et al in 2017. It connects each layer with others in the way of feedforward. The structure of DenseNet is shown in Figure 4.

In Figure 4, $x_i$ is the feature map of the output, while $H_i$ represents the transport layer. There are $l(l+1)/2$ connections in the network with $l$ layers. Each layer is connected to all the other layers; thus, each layer can receive all the feature maps of the preceding layers. The feature map of each layer can be expressed in Equation (7):

$$x_l = H_l[x_0, x_1, \ldots x_{l-1}]. \tag{7}$$

The structure of DenseNet makes it easy to alleviate gradient fading. In addition, DenseNet can also enhance feature transmitting and reduce the number of parameters to a certain extent. The structure of our proposed densely connected network is described in detail in Section 3.4.

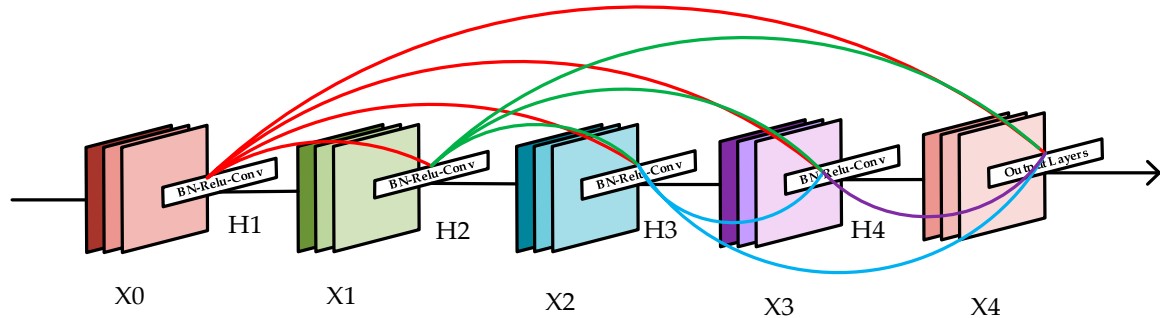

**Figure 4.** The structure of DenseNet.

### 3.3. Multi-Scale Detecting Layers

Three scales of detecting layers are used in YOLO-V3 to detect multi-scale targets. Among them, the scale with a feature map down-sampled by 32× is responsible for detecting the big targets. The scale with a feature map down-sampled by 16× is responsible for detecting the medium-sized targets, and the scale with a feature map down-sampled by 8× is responsible for detecting the small targets. The remote sensing images contain a large number of small targets. In order to get more fine-grained features and more detailed location information, the 4th scale with a feature map down-sampled by 4× is added to the network as a new detecting layer.

### 3.4. Our Model

From what has been discussed above, the proposed MRFF-YOLO adopted ResNet, Res2Net, DenseNet, and multi-scales detecting layers. The structure of MRFF-YOLO is shown in Figure 5.

MRFF-YOLO adopts 'Res2 blocks' to replace residual blocks in YOLO-V3. 'Res2 block' contains several 'Res2 units' (Figure 3), while 'residual block' contains several 'residual units' (Figure 2). The parameter settings of the feature extraction network of YOLO-V3 and MRFF-YOLO are shown in Tables 1 and 2.

Since each $x_i$ in 'Res2 block' is in the same size but contains only $1/N$ number of channels compared with 'RES Block', the number of parameters of the network has not been increased.

The structure of the Dense blocks in Figure 5 is shown in Figure 6.

**Table 1.** The parameter settings of the feature extraction network of YOLO-V3.

| | YOLO-V3 | |
|---|---|---|
| | Conv($3 \times 3/2 \times 32$)-BN-ReLU | Convolutional |
| | Conv($3 \times 3/2 \times 64$)-BN-ReLU | Convolutional |
| RES × 1 | Conv($1 \times 1 \times 32$)-BN-ReLU-Conv($3 \times 3 \times 64$)-BN-ReLU | Residual |
| | Conv($3 \times 3/2 \times 128$)-BN-ReLU | Convolutional |
| RES × 2 | Conv($1 \times 1 \times 64$)-BN-ReLU-Conv($3 \times 3 \times 128$)-BN- ReLU | Residual |
| | Conv($3 \times 3/2 \times 256$)-BN-ReLU | Convolutional |
| RES × 8 | Conv($1 \times 1 \times 128$)-BN-ReLU-Conv($3 \times 3 \times 256$)-BN-ReLU | Residual |
| | Conv($3 \times 3/2 \times 512$)-BN- ReLU | Convolutional |
| RES × 8 | Conv($1 \times 1 \times 256$)-BN-ReLU-Conv($3 \times 3/ \times 512$)-BN-ReLU | Residual |
| | Conv($3 \times 3/2 \times 1024$)-BN-ReLU | Convolutional |
| RES × 4 | Conv($1 \times 1 \times 512$)-BN-ReLU-Conv($3 \times 3 \times 1024$)-BN-ReLU | Residual |

**Table 2.** The parameter setting of the feature extraction network of Multi-Receptive Fields Fusion YOLO (MRFF-YOLO).

| | MRFF-YOLO | |
|---|---|---|
| | Conv($3 \times 3/2 \times 32$)-BN-ReLU | Convolutional |
| | Conv($3 \times 3/2 \times 64$)-BN-ReLU | Convolutional |
| RES $\times$ 1 | Conv($1 \times 1 \times 32$)-BN-ReLU-Conv($3 \times 3 \times 64$)-BN-ReLU | Residual |
| | Conv($3 \times 3/2 \times 128$)-BN-ReLU | Convolutional |
| RES $\times$ 2 | Conv($1 \times 1 \times 64$)-BN-ReLU-Conv($3 \times 3 \times 128$)-BN-ReLU | Residual |
| | Conv($3 \times 3/2 \times 256$)-BN-ReLU | Convolutional |
| RES $\times$ 8 | Conv($1 \times 1 \times 128$)-BN-ReLU-Conv($3 \times 3 \times 256$)-BN-ReLU | Residual |
| | Conv($3 \times 3/2 \times 512$)-BN-ReLU | Convolutional |
| RES2 $\times$ 8 | Conv($1 \times 1 \times 256$)-BN-ReLU | Residual |
| | $x_1$ <br> $x_2$: Conv($3 \times 3 \times 128$)-BN-ReLU <br> $x_3$: Conv($3 \times 3 \times 128$)-BN-ReLU <br> $x_4$: Conv($3 \times 3 \times 128$)-BN-ReLU | Concat |
| | Conv($3 \times 3/2 \times 1024$)-BN-ReLU | Convolutional |
| RES2 $\times$ 4 | Conv($1 \times 1 \times 512$)-BN-ReLU | Residual |
| | $x_1$ <br> $x_2$: Conv($3 \times 3 \times 256$)-BN-ReLU <br> $x_3$: Conv($3 \times 3 \times 256$)-BN-ReLU <br> $x_4s$: Conv($3 \times 3 \times 256$)-BN-ReLU | Concat |

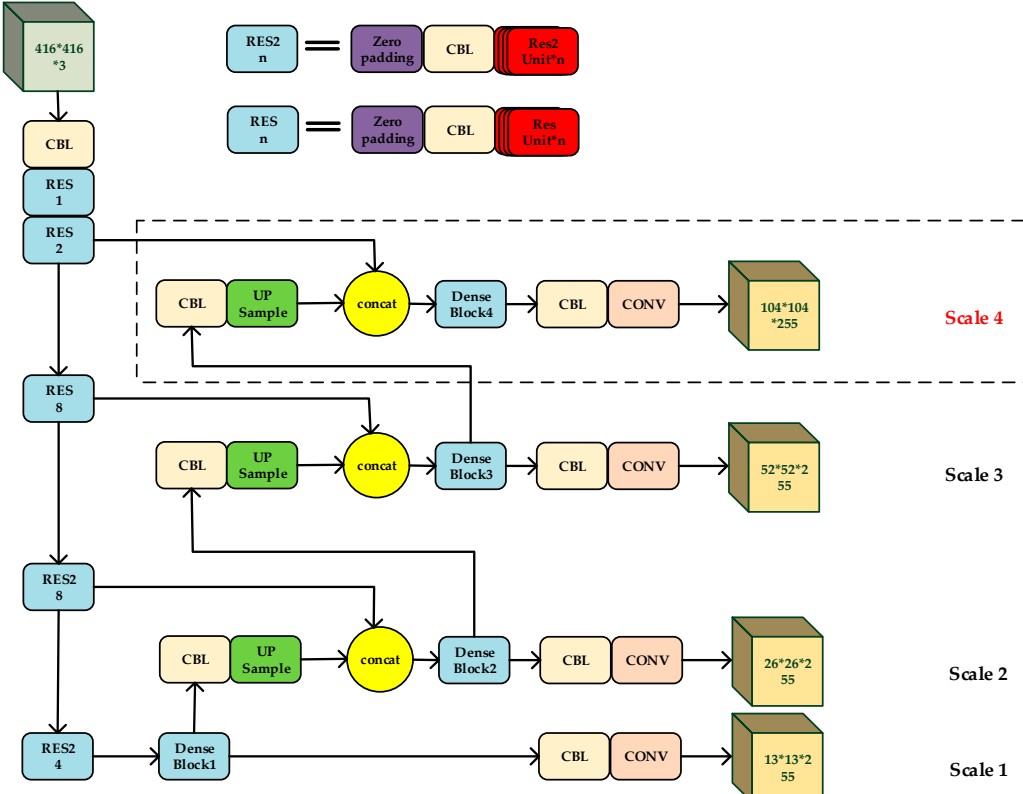

**Figure 5.** The structure of the proposed network.

As shown in Figure 6, $H_0$ represents the convolutional layer. $H_1$–$H_4$ represent the transport layers: Conv ($1 \times 1 \times$ M) $-$ BN $-$ ReLU $-$ Conv ($3 \times 3 \times$ 2M) $-$ BN $-$ ReLU. The increments of the feature maps for each layer of 'Dense block 1' to 'Dense block 4' are 128, 64, 32, and 16, respectively. Compared with

the convolution layer, the proposed 'Dense block' can alleviate the gradient fading while improving the depth of the network. At the same time, the parameters of the network are greatly reduced.

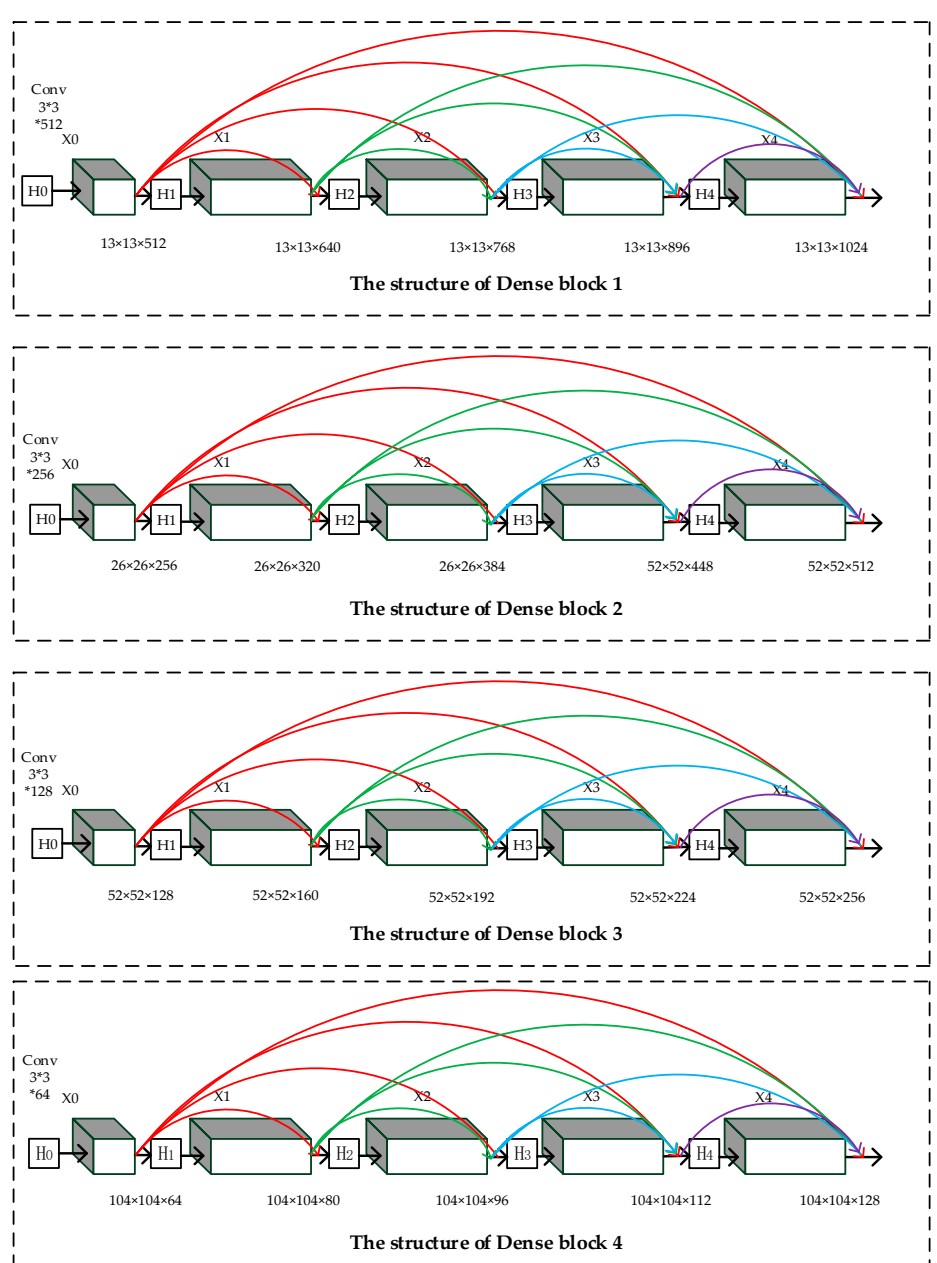

**Figure 6.** The structure of Dense blocks.

### 3.5. K-Means for Anchor Boxes

Anchor box is used to detect multiple targets in one grid cell, which was a concept firstly proposed in Faster-RCNN. Inspired by Faster-RCNN, YOLO-V3 adopts anchor boxes to match the length-to-width ratios of targets better. Different from Faster-RCNN, which sets the sizes of anchor boxes manually, we executes K-means on the dataset to acquire anchor boxes in advance for YOLO-V3. The K-means function conducts latitude clustering to make the anchor boxes and adjacent ground truth as approximate as possible, which means they can have larger IOU values. For each ground truth, $gt_j(x_j, y_j, w_j, h_j)$, $j \in \{1, \dots N\}$, and $(x_j, y_j)$ represent the center of the ground truth, while $(w_j, h_j)$ refers to the height and the width of the ground truth. The distance between the ground truth and bounding box is defined as follows [50]:

$$d(\text{box} - \text{centroid}) = 1 - \text{IOU}(\text{box}, \text{centroid}). \tag{8}$$

IOU represents the intersection over union, which is defined in Equation (9):

$$\text{IOU} = \frac{S_{overlap}}{S_{union}}. \tag{9}$$

The larger the value of IOU between the ground truth and bounding box, the smaller the distance will be. The steps of the algorithm are shown in Table 3.

**Table 3.** K-means for anchor boxes.

| The K-Means Clustering for Anchor Boxes |
| --- |
| 1: Set $k$ random cluster center points: $(W_i, H_i)$, $i \in \{1, \ldots, k\}$. $W_i, H_i$ represent the width and height of each anchor box. |
| 2: Then, we calculated the distance between each ground truth and each cluster center: $d(box - centroid) = 1 - \text{IOU}(box, centroid)$. Since the position of the anchor box is not fixed, the center point of each ground truth is coincident with the clustering center. |
| 3: Recalculate the cluster center for each cluster: $W_i' = 1/N_i \sum w_i$, $H_i' = 1/N_i \sum h_i$. |
| 4: Repeat step 2 and step 3 until the clusters converge. |

### 3.6. Decoding Process

In order to get the final bounding boxes, we need to decode the predicted value. The relationship between the bounding box and its corresponding prediction box is shown in Figure 7. In Figure 7, $t_x, t_y, t_w, t_h$ refer to the predicted values, while $c_x, c_y$ represent the offset of the grid relative to the upper left. The location and size information of the bounding box are shown in Equation (10).

$$\begin{aligned}
b_x &= \sigma(t_x) + c_x \\
b_y &= \sigma(t_y) + c_y \\
b_w &= p_w e^{t_w} \\
b_h &= p_h e^{t_h} \\
c_x &= 1/(1 + e^{-x})
\end{aligned} \tag{10}$$

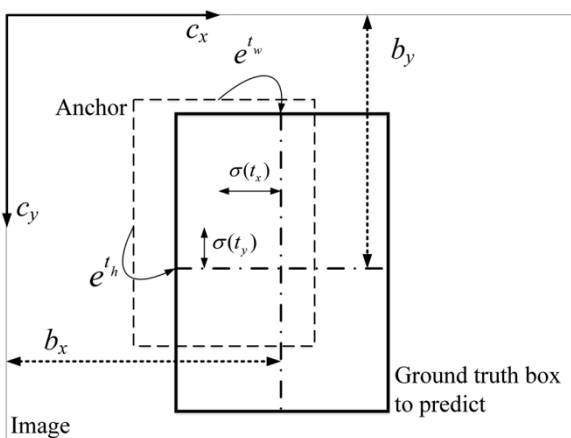

**Figure 7.** The decoding schematic.

### 3.7. Remove Redundant Bounding Boxes

After decoding, the network will generate the bounding boxes of the targets. In order to eliminate redundant bounding boxes that correspond to the same targets, we run Non-Maximum Suppression (NMS) on bounding boxes. The NMS algorithm contains three steps:

① Step 1: For the bounding boxes with the same category, we compare the value of IOU between every bounding box with others.

② Step 2: If the value of IOU is larger than the threshold, then we shall consider that they correspond to the same target, and the bounding box with higher confidence will be retained.

③ Step 3: Repeat step 1 and step 2 until all the boxes are retained.

Algorithm 1 exhibits the detailed steps of NMS for our approach:

---

**Algorithm 1** The pseudocode of NMS

---

Original Bounding Boxes:

　　　$B = [b_1, \ldots b_s]$, $C = [c_1, \ldots c_s]$, $threshold = 0.6$

　　　$B$ refers to the list of the bounding boxes generated by the network

　　　$C$ refers to the list of the confidences corresponding to the bounding boxes in $C$

Detection result:

　　　$F$ refers to the list of the final bounding boxes

1:　　　$F \leftarrow []$

2:　　　while $B \neq []$ do:

3:　　　　　$k \leftarrow \text{argmax} c$

4:　　　　　$F \leftarrow F.append(b_k)$; $B \leftarrow delB[b_k]$; $C \leftarrow delC[b_k]$

5:　　　　　**for** $b_i \in B$ do:

6:　　　　　　**if** $IOU(b_i, b_i) \geq thresold$

7:　　　　　　　$B \leftarrow delB[b_k]$; $C \leftarrow delC[b_k]$

8:　　　　　**end**

9:　　　**end**

10:　**end**

---

## 4. Results

In this section, we conduct experiments on RSOD and UCS-AOD datasets and compared our approach with other state-of-the-art target detection models such as YOLO-V2, YOLO-V3, SSD, Faster-RCNN, etc. The experimental conditions are shown as follows: Framework: Python3.6.5, tensorflow-GPU1.13.1. Operating system: Windows 10. CPU: i7-7700k. GPU: NVIDIA GeForce RTX 2070. 50,000 training steps were set. The learning rate of our approach decreased from 0.001 to 0.0001 after 30,000 steps and to 0.00001 after 40,000 steps. The initialization parameters are displayed in Table 4.

**Table 4.** The initialization parameters of training.

| Input Size | Batch Size | Momentum | Learning Rate | Training Step |
|:---:|:---:|:---:|:---:|:---:|
| 416 × 416 | 8 | 0.9 | 0.001–0.00001 | 50,000 |

### 4.1. Anchor Boxes of Our Model

We run K-means on the RSOD and UCS-AOD datasets to get anchor boxes. In Figure 8, we can see the average IOU with different numbers of clusters. The curves of IOU become more and more flat as the number of clusters increase. Since there are four detecting layers in the network of our approach, we select 12 clusters (anchor boxes) and three anchor boxes for each detection scale. The sizes of the anchor boxes for the RSOD dataset are as follows: (21, 24), (25, 31), (33, 41), (51, 54), (61, 88), (82, 91), (109, 114), (121, 153), (169, 173), (232, 214), (241, 203), and (259, 271). Among them, (21, 24), (25, 31), and (33, 41) are the anchor boxes for Scale 4; (51, 54), (61, 88), and (82, 91) are the anchor boxes for Scale 3; (109, 114), (121, 153), and (169, 173) are the anchor boxes for Scale 2; and (232, 214), (241, 203), and (259, 271) are the anchor boxes for Scale 1. The sizes of the anchor boxes for the UCS-AOD dataset are as follows: (19, 22), (23, 29), (31, 38), (49, 52), (63, 86), (80, 92), (101, 124), (118, 147), (152, 167), (225, 201), (231, 212), and (268, 279). Among them, (19, 22), (23, 29), and (31, 38) are the anchor boxes for Scale 4; (49, 52), (63, 86), and (80, 92) are the anchor boxes for Scale 3; (101, 124), (118, 147), and (152, 167) are the anchor boxes for Scale 2, and (225, 201), (231, 212), and (268, 279) are the anchor boxes for Scale 1.

The sizes of the corresponding anchor boxes for the RSOD dataset and UCS-AOD dataset are shown in Table 5.

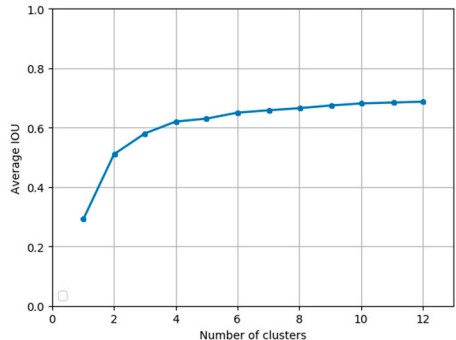

The curve of the RSOD dataset

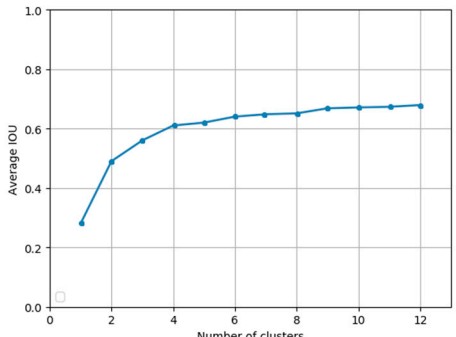

The curve of the UCS-AOD dataset

**Figure 8.** The relationship between number of clusters and average IOU (Intersection-Over-Union) by K-means clustering.

**Table 5.** The corresponding anchor boxes for the RSOD and UCS-AOD datasets.

| Dataset | | RSOD | UCS-AOD |
|---|---|---|---|
| | Scale 1 | (232, 214), (241, 203), (259, 271) | (225, 201), (231, 212), (268, 279) |
| Anchors | Scale 2 | (109, 114), (121, 153), (169, 173) | (101, 124), (118, 147), (152, 167) |
| | Scale 3 | (51, 54), (61, 88), (82, 91) | (49, 52), (63, 86), (80, 92) |
| | Scale 4 | (21, 24), (25, 31), (33, 41) | (19, 22), (23, 29), (31, 38) |

*4.2. The Evaluation Indicators*

To evaluate a binary classification model, we can divide all the results in four categories: True Positive (TP), False Positive (FP), True Negative (TN), and False Negative (FN). We exhibit the confusion matrix in Table 6:

**Table 6.** The confusion matrix [51].

| Actual | Predicted | Confusion Matrix |
|---|---|---|
| Positive | Positive | TP |
| Negative | Positive | FP |
| Positive | Negative | FN |
| Negative | Negative | TN |

As shown in Table 6, TP denotes the sample that is positive in actuality and positive in prediction; FP denotes the sample that is negative in actuality but positive in prediction; FN denotes the sample that is positive in actuality but negative in prediction; TN denotes the sample that is negative in actuality and negative in prediction. With the confusion matrix, precision and recall are defined in Equations (11) and (12):

$$Precision = \frac{TP}{TP + FP} \tag{11}$$

$$Recall = \frac{TP}{TP + FN}. \tag{12}$$

Accuracy and recall are two indicators that check and balance each other. The tradeoff between them is hard. With aiming at measuring the precision of detecting targets with different categories, average precision (AP) and mean average precision (mAP) are introduced, which are the most important evaluation indicators of target detection.

Average precision (AP) is defined as:

$$AP_i = \int_0^1 P_i(R_i).dR_i \tag{13}$$

where $P_i$ refers to the precision of the $i-$th category, $R_i$ refers to the recall of the $i-$th category. $P_i(R_i)$ is the function with $R_i$ as its independent variable and $P_i$ as its dependent variable. It measures the performance of target detection for a certain category.

The mean average precision (mAP) is defined as:

$$mAP = \frac{\sum\limits_{i=1}^{c} AP_i}{c}. \tag{14}$$

It measures the performance of target detection for all the $c$ categories.

In addition, FPS is also an important indicator for target detection for measuring the real-time performance. It refers to the number of frames processed by the target detection algorithm in one second.

### 4.3. Experimental Process and Analysis

To evaluate the validity of our approach for remote sensing target detection, we selected RSOD and UCS-AOD as our experimental datasets. Generally speaking, if the ground truth of the target takes up less than 0.12% pixels of the whole image, we divided it into the category of small targets. If the ground truth of the target takes up 0.12–0.5% pixels of the whole image, we divided it into the category of medium targets. Otherwise, if the ground truth of the target takes up more than 0.5% pixels of the whole image, we divided it into the category of large targets. The RSOD dataset contains a mass of aerial images. The targets marked in the samples are divided into four categories, including aircraft, playground, oil tank, and overpass. Among them, most of the targets of the aircraft and oil tank are small or medium in size, and the size of the playgrounds and overpasses are large. In addition to scale diversity, the samples are also obtained under different light conditions and backgrounds of varying degrees of complexity. UCS-AOD is the dataset of target detection in aerial images. Tables 7 and 8 contain statistics tables of the datasets.

**Table 7.** Statistics table of RSOD.

| Dataset | Class | Image | Instances | Target Amount | | |
|---|---|---|---|---|---|---|
| | | | | Small | Medium | Large |
| Training set | Aircraft | 446 | 4993 | 3714 | 833 | 446 |
| | Oil tank | 165 | 1586 | 724 | 713 | 149 |
| | Overpass | 176 | 180 | 0 | 0 | 180 |
| | Playground | 189 | 191 | 0 | 12 | 179 |
| Test set | Aircraft | 176 | 1257 | 741 | 359 | 157 |
| | Oil tank | 63 | 567 | 257 | 213 | 97 |
| | Overpass | 36 | 41 | 0 | 0 | 41 |
| | Playground | 49 | 52 | 0 | 0 | 52 |

**Table 8.** UCS-AOD dataset statistics.

| Dataset | Class | Image | Instances |
|---|---|---|---|
| Training set | Aircraft | 600 | 3591 |
| | Car | 310 | 4475 |
| Test set | Aircraft | 400 | 3891 |
| | Car | 200 | 2639 |

Figure 9 exhibits some of the samples in RSOD and UCS-AOD; it shows the targets under different conditions. There are the samples under strong light condition, the samples under weak light condition, and the samples with complex background condition.

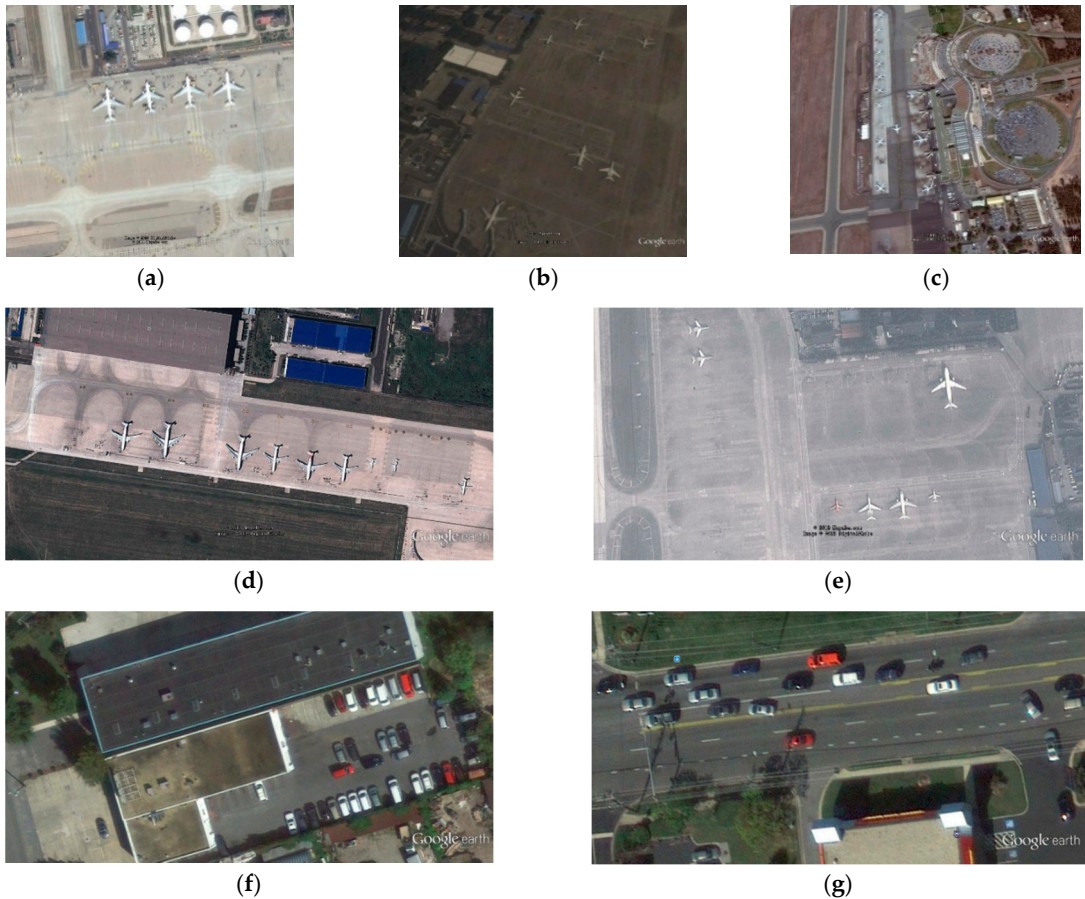

(**a**)                              (**b**)                              (**c**)

(**d**)                              (**e**)

(**f**)                              (**g**)

**Figure 9.** The samples of the dataset. (**a**) Aircraft and oiltank targets (**b**) Small aircraft targets under weak light condition (**c**) Small aircraft targets under complex background. (**d**) Small aircraft targets. (**e**) Small aircraft targets under bad weather condition. (**f**) Densely contributed car targets (**g**) Car targets.

4.3.1. Experimental Results and Comparative Analysis

Three evaluation indexes are adopted to verify the performance of our approach. They are the mAP, Frames Per Second (FPS), and leak detection rate, respectively. We compared the performance of our approach with the state-of-the-art target detection algorithms in the RSOD dataset. The contrastive results are shown in Table 9. In addition, if we differentiate targets by size, the contrastive results are shown in Table 10.

Table 9 demonstrates that the proposed MRFF-YOLO is superior to the other state-of-the-art target detections in mAP. FPS did not reduce much compared with YOLO-V3. The mAP of MRFF-YOLO for remote sensing target detection is 88.33%, which increases by 11.23%, 10.54%, and 11.75% compared with YOLO-V3, UAV-YOLO, and DC-SPP-YOLO, respectively. In addition, the accuracy of detecting small and medium targets such as aircrafts and oil tanks has been significantly improved. In terms of detection speed, MRFF-YOLO satisfies the real-time performance of remote sensing targets. Experimental results indicated that the improved MRFF-YOLO can obviously improve the accuracy of detecting remote sensing targets under complex background. Not only that, MRFF-YOLO can meet the demand of real-time detection. In particular, the detection effect of small targets is more advantageous. Table 10 shows the contrastive results of different sizes. We can see that MRFF-YOLO is superior to YOLO-V3 in detecting small targets.

**Table 9.** The contrastive results of different categories in the RSOD dataset.

| Method | Backbone | AP (%) | | | | | FPS |
|---|---|---|---|---|---|---|---|
| | | Aircraft | Oil Tank | Overpass | Playground | mAP (IOU = 0.5) | |
| Faster RCNN | VGG-16 | 85.85 | 86.67 | 88.15 | 90.35 | 87.76 | 6.7 |
| SSD | VGG-16 | 69.17 | 71.20 | 70.23 | 81.26 | 72.97 | 62.2 |
| DSSD [52] | ResNet-101 | 72.12 | 72.49 | 72.10 | 83.56 | 75.07 | 6.1 |
| ESSD [53] | VGG-16 | 73.08 | 72.94 | 73.61 | 84.27 | 75.98 | 37.3 |
| FFSSD [54] | VGG-16 | 72.95 | 73.24 | 73.17 | 84.08 | 75.86 | 38.2 |
| YOLO | GoogleNet | 52.71 | 49.58 | 51.06 | 62.17 | 53.88 | 61.4 |
| YOLO-V2 | DarkNet19 | 62.35 | 67.74 | 68.38 | 78.51 | 69.25 | 35.6 |
| YOLO-V3 | DarkNet53 | 74.30 | 73.85 | 75.08 | 85.16 | 77.10 | 29.7 |
| YOLO-V3 tiny | DarkNet19 | 54.14 | 56.21 | 59.28 | 64.20 | 58.46 | 69.8 |
| UAV-YOLO [41] | Figure 1 in [41] | 74.68 | 74.20 | 76.32 | 85.96 | 77.79 | 30.12 |
| DC-SPP-YOLO [55] | Figure 5 in [55] | 73.16 | 73.52 | 74.82 | 84.82 | 76.58 | 33.5 |
| MRFF-YOLO | (Table 2) | 87.16 | 86.56 | 87.56 | 92.05 | 88.33 | 25.1 |

**Table 10.** The contrastive results of different sizes in the RSOD dataset.

| Method | Backbone | AP (%) | | | Leak Detection Rate (%) |
|---|---|---|---|---|---|
| | | Small | Medium | Large | |
| Faster RCNN | VGG-16 | 84.73 | 87.87 | 89.18 | 11.8 |
| SSD | VGG-16 | 70.38 | 73.41 | 77.51 | 21.1 |
| DSSD [52] | ResNet-101 | 74.42 | 75.18 | 77.70 | 15.2 |
| ESSD [53] | VGG-16 | 75.12 | 75.84 | 78.12 | 16.5 |
| FFSSD [54] | VGG-16 | 72.62 | 74.78 | 82.56 | 18.2 |
| YOLO | GoogleNet | 52.25 | 51.68 | 60.35 | 33.6 |
| YOLO-V2 | DarkNet19 | 63.20 | 68.53 | 69.28 | 24.3 |
| YOLO-V3 | DarkNet53 | 74.52 | 75.63 | 76.14 | 19.5 |
| YOLO-V3 tiny | DarkNet19 | 55.26 | 56.47 | 60.17 | 31.4 |
| UAV-YOLO [41] | Figure 1 in [41] | 75.45 | 75.15 | 76.85 | 17.1 |
| DC-SPP-YOLO [55] | Figure 5 in [55] | 75.41 | 74.67 | 76.41 | 15.9 |
| MRFF-YOLO | (Table 2) | 87.76 | 88.42 | 91.85 | 8.5 |

For the universality of the performance of MRFF-YOLO for remote sensing target detection, we chose another dataset, UCS-AOD, as an additional verification. The contrastive results lie in Table 11. We can see from Tables 10 and 11 that the leak detection rate of MRFF-YOLO is prominently lower than the original YOLO-V3 and other classical target detection algorithms.

**Table 11.** The contrastive results in the UCS-AOD dataset.

| Method | Backbone | AP (%) | | Leak Detection Rate (%) | mAP (IOU = 0.5) | FPS |
|---|---|---|---|---|---|---|
| | | Aircraft | Car | | | |
| Faster RCNN | VGG-16 | 87.31 | 86.48 | 13.8 | 86.90 | 6.1 |
| SSD | VGG-16 | 70.24 | 72.61 | 23.7 | 71.43 | 61.5 |
| DSSD [52] | ResNet-101 | 73.17 | 74.19 | 16.1 | 73.68 | 5.2 |
| ESSD [53] | VGG-16 | 73.62 | 75.06 | 15.9 | 74.34 | 33.2 |
| FFSSD [54] | VGG-16 | 71.15 | 74.63 | 17.6 | 72.89 | 34.6 |
| YOLO | GoogleNet | 54.57 | 57.70 | 47.6 | 56.14 | 64.2 |
| YOLO-V2 | DarkNet19 | 63.17 | 68.42 | 23.0 | 65.80 | 34.3 |
| YOLO-V3 | DarkNet53 | 75.71 | 75.62 | 18.5 | 75.67 | 27.6 |
| YOLO-V3 tiny | DarkNet19 | 57.58 | 56.35 | 35.2 | 56.97 | 65.3 |
| UAV-YOLO [41] | Figure 1 in [41] | 75.12 | 75.60 | 16.5 | 75.36 | 28.4 |
| DC-SPP-YOLO [55] | Figure 5 in [55] | 76.52 | 74.61 | 17.4 | 75.57 | 30.4 |
| MRFF-YOLO | (Table 2) | 91.23 | 90.28 | 9.1 | 90.76 | 24.3 |

Figure 10 exhibits some of the detection results of our proposed MRFF-YOLO. A set of 19 samples contains the remote sensing targets with different sizes and categories. They are under backgrounds of varying degrees of complexity and in different light conditions. The angles of view from which images are acquired are also quite different. Each target is shown in Figure 10, which certified the excellent performance of our approach for remote sensing target detection.

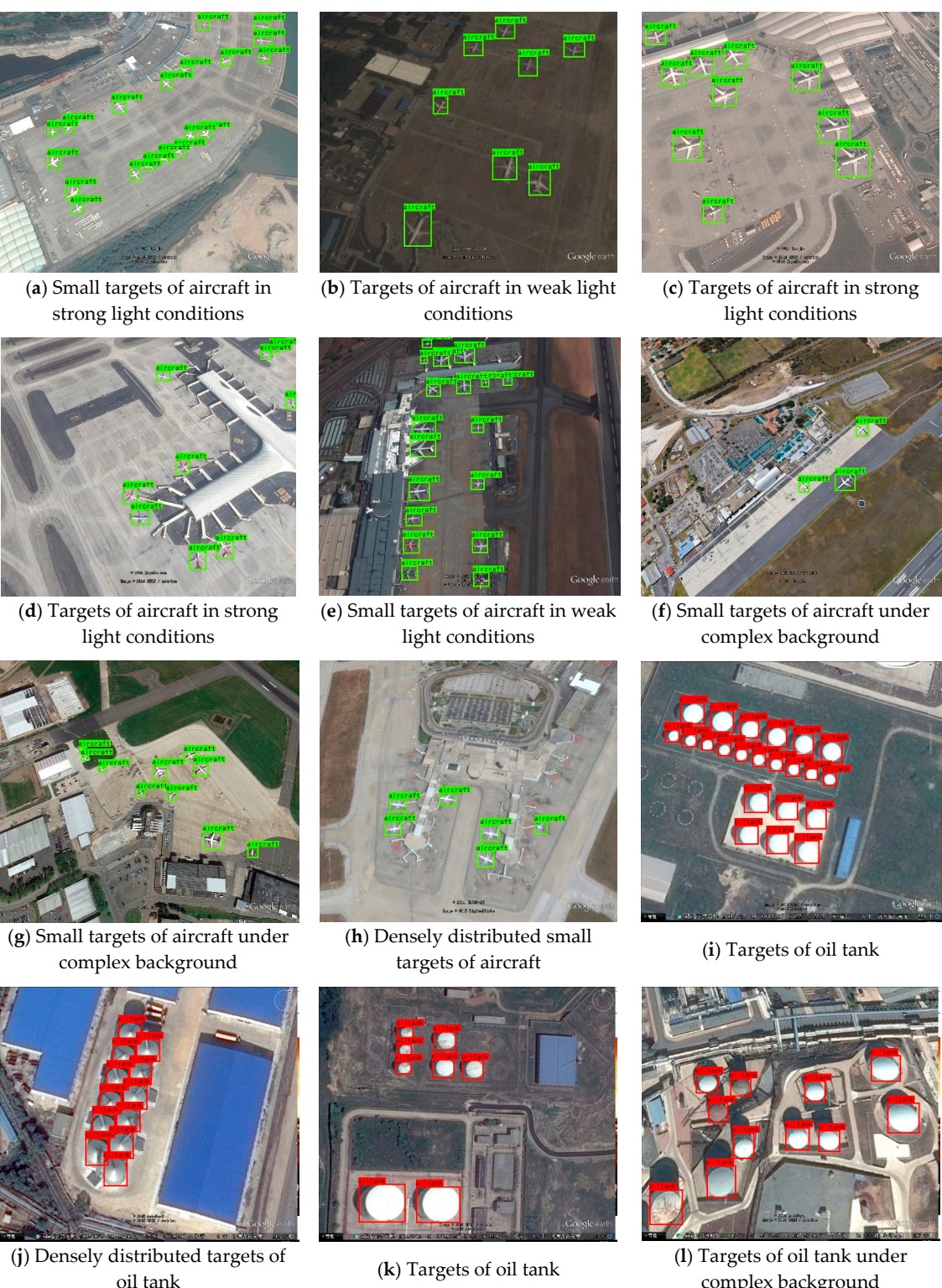

(**a**) Small targets of aircraft in strong light conditions

(**b**) Targets of aircraft in weak light conditions

(**c**) Targets of aircraft in strong light conditions

(**d**) Targets of aircraft in strong light conditions

(**e**) Small targets of aircraft in weak light conditions

(**f**) Small targets of aircraft under complex background

(**g**) Small targets of aircraft under complex background

(**h**) Densely distributed small targets of aircraft

(**i**) Targets of oil tank

(**j**) Densely distributed targets of oil tank

(**k**) Targets of oil tank

(**l**) Targets of oil tank under complex background

**Figure 10.** *Cont.*

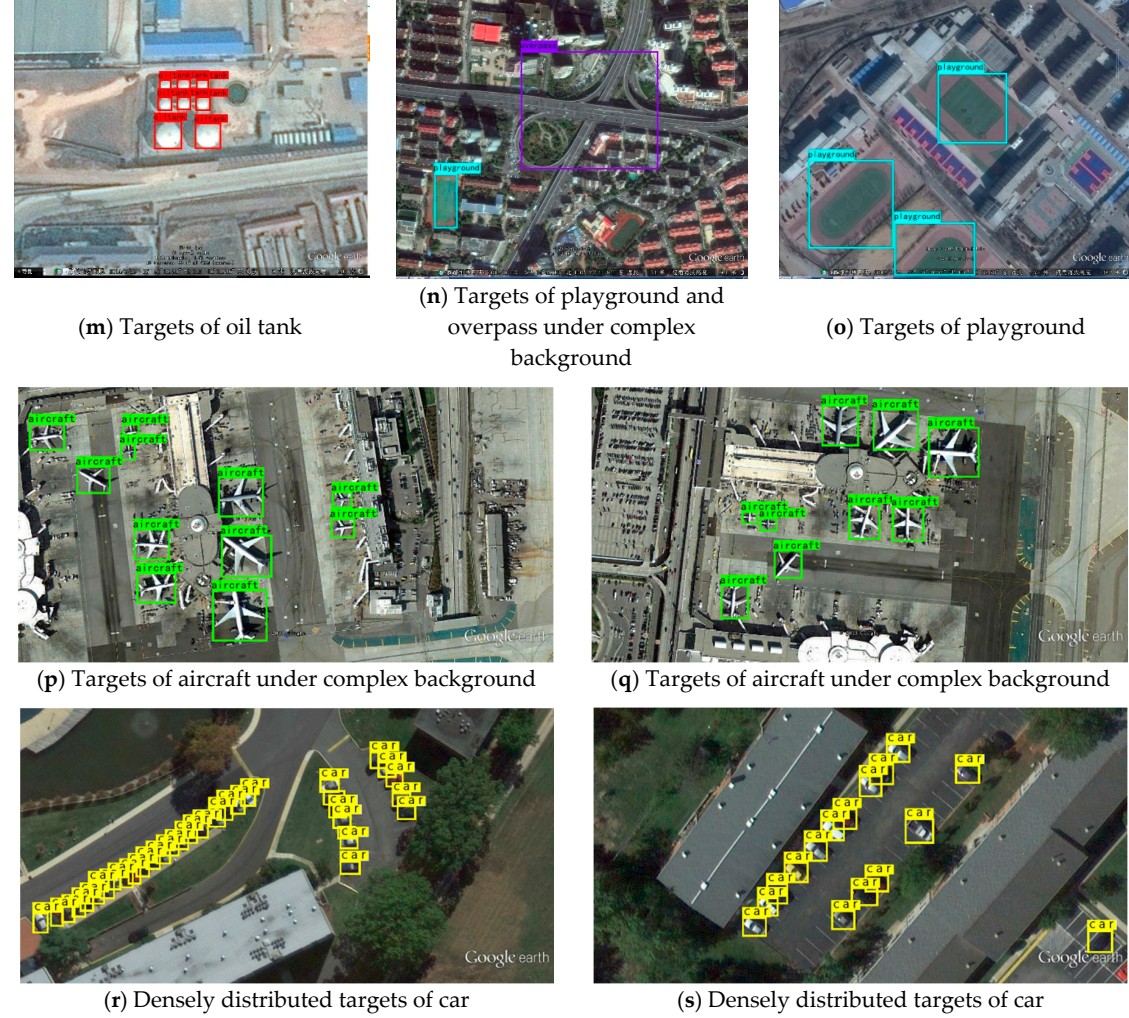

(**m**) Targets of oil tank

(**n**) Targets of playground and overpass under complex background

(**o**) Targets of playground

(**p**) Targets of aircraft under complex background

(**q**) Targets of aircraft under complex background

(**r**) Densely distributed targets of car

(**s**) Densely distributed targets of car

**Figure 10.** The detection results of the improved YOLO-V3.

### 4.3.2. Ablation Experiments

Section 4.3.1 has proved the advantage of MRFF-YOLO. In order to analyze the impact of 'Dense block', 'Res2 block', and the 4th detection layer on mAP and FPS, different module combination modes were set up in the experiment, and the RSOD dataset was chosen.

With the aiming at verifying the validity of 'Res2 block' in a feature extraction network and the 4th detection, different module combination modes are set in the experiment. The experimental results are shown in Tables 12 and 13. Among them, Table 12 exhibits the ablation experimental result with three detection layers, while Table 13 exhibits the ablation experimental result with four detection layers. The detection layers are the same as those of the original YOLO-V3.

**Table 12.** The influence of 'Res2 block' on remote sensing target detection with three detection layers.

| | RES2 8 | RES2 4 | AP (%) | | | | | FPS |
|---|---|---|---|---|---|---|---|---|
| | | | Aircraft | Oil tank | Overpass | Playground | mAP (IOU = 0.5) | |
| 1 | | | 74.30 | 73.85 | 75.08 | 85.16 | 77.10 | 29.7 |
| 2 | √ | | 75.05 | 74.37 | 75.61 | 85.82 | 77.71 | 30.2 |
| 3 | | √ | 74.86 | 74.12 | 75.85 | 85.73 | 77.64 | 30.1 |
| 4 | √ | √ | 75.21 | 74.86 | 76.27 | 86.12 | 78.12 | 31.5 |

**Table 13.** The influence of 'Res2 block' on remote sensing target detection with four detection layers.

| | RES2 8 | RES2 4 | AP (%) | | | | | FPS |
|---|---|---|---|---|---|---|---|---|
| | | | Aircraft | Oil Tank | Overpass | Playground | mAP (IOU = 0.5) | |
| 1 | | | 84.72 | 84.81 | 85.07 | 90.41 | 86.25 | 22.8 |
| 2 | √ | | 85.31 | 85.26 | 85.27 | 90.81 | 86.66 | 23.3 |
| 3 | | √ | 85.58 | 85.39 | 85.12 | 90.52 | 86.65 | 23.3 |
| 4 | √ | √ | 86.51 | 85.71 | 86.16 | 91.57 | 87.49 | 23.5 |

The results of the contrast in Tables 13 and 14 show that with 'Res2 block' in the feature extraction network, the mAP improved from 77.10% to 78.12% and from 86.25% to 87.49%, respectively. In addition, the detection speed improved from 29.7 to 31.5 FPS and from 22.8 to 23.5 FPS, respectively. The experimental contrast certified the effectiveness of the improvement in the feature extraction network. In order to verify the impact of an additional detection layer on the detection accuracy, we compared the 1st experiment to the 4th experiment in Table 12 with those in Table 13, respectively. The mAP improved by 9.15%, 8.95%, 9.01%, and 9.37%, respectively. For smaller targets such as aircraft, the accuracy improved more obviously, which proves that the additional detection layer is suitable for smaller remote sensing target detection.

**Table 14.** The influence of 'Dense block' on remote sensing target detection.

| mAP | FPS | Dense Block 1 | Dense Block 2 | Dense Block 3 | Dense Block 4 |
|---|---|---|---|---|---|
| 87.49 | 23.5 | | | | |
| 87.54 | 23.8 | | | | √ |
| 87.69 | 24.3 | | | √ | √ |
| 88.13 | 24.8 | | √ | √ | √ |
| 88.33 | 25.1 | √ | √ | √ | √ |

The ablation experiments demonstrated in Tables 12–14 indicated that each module we proposed is efficient for improving the accuracy of remote sensing target detection. Among them, the proposed 'Res2 blocks' in the feature extraction network and 'Dense block 1' to 'Dense block 4' in the detection layers not only improved the accuracy but also sped up the detection. In addition, the 4th detection layer improved the performance of detecting small targets in a large degree at the expense of some detection speed. Generally speaking, MRFF-YOLO is an excellent model for real-time remote sensing target detection.

Table 14 compares the experimental effects of each 'Dense block' in Figure 6. With 'Dense block 1' to 'Dense block 4' added in the detection layers, the mAP improved from 87.49% to 88.33% and the FPS improved from 23.5 to 25.1, which indicated that the Dense blocks we proposed in the detection layers can modestly improve the accuracy of detecting remote sensing targets and accelerate the velocity of detection simultaneously.

### 4.3.3. Comparison of Detection Effect

Figure 10 shows the perfect performance of MRFF-YOLO for remote sensing target detection. Besides the detection results in Figure 10, the comparison of detection effect between MRFF-YOLO and YOLO-V3 is also provided. The RSOD dataset contains a mass of small targets, so we chose its detection results for comparison. In Figure 11, 10 samples were chosen for comparison.

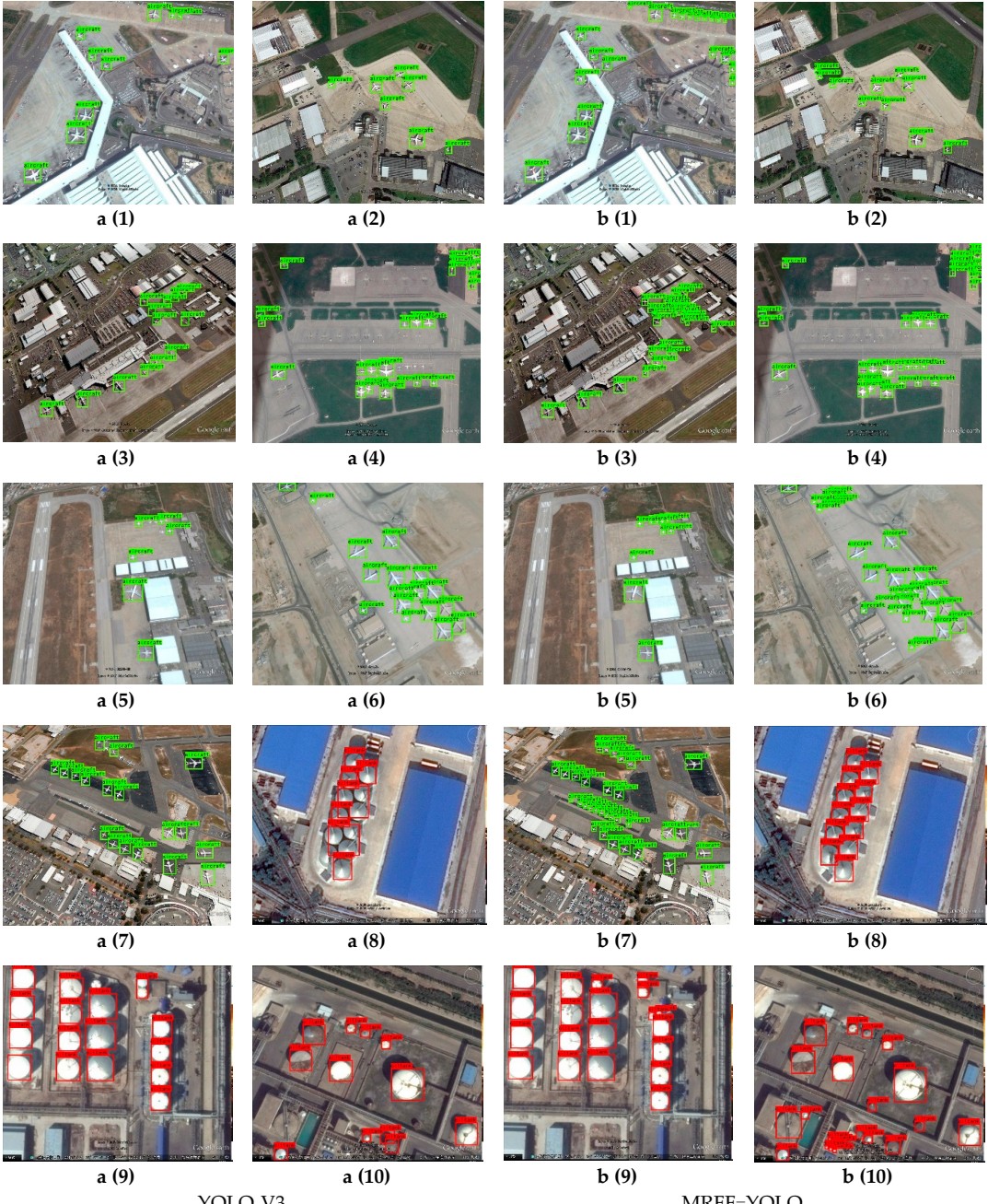

**Figure 11.** The comparison results of YOLO-V3 and MRFF-YOLO: (**a1**)–(**a10**) the detection results of YOLO-V3; (**b1**)–(**b10**) the detection results of MRFF-YOLO.

Figure 11 provided 20 images of 10 sets to compare the detection effect of YOLO-V3 and MRFF-YOLO intuitively. The images contain a mass of densely distributed targets that are small or medium in size. Among them, the 1st list and the 2nd list are the images detected by YOLO-V3, while the 3rd list and the 4th list are the images detected by MRFF-YOLO. Figure 11 clearly showed that several targets were not detected or erratically detected by YOLO-V3. Especially if the targets are small and densely distributed, there will be situations in which YOLO-V3 may predict several targets as one (a (3), a (7), a (8), a (9)) or judge shadows as targets (a (10)). On the other hand, all the marked targets were detected faultlessly. The contrast experiment in this section showed that our improved YOLO-V3, MRFF-YOLO, can detect densely distributed small and medium remote sensing targets better than original YOLO-V3.

## 5. Conclusions

Aimed at the characteristics of remote sensing targets for which a large number of small targets exist in remote sensing images and their distribution is relatively dense, a series of improvements were proposed based on YOLO-V3. In order to realize the multi-scale feature extraction of the target, Res2Net was adopted to improve the capability of the feature extraction network. Posteriorly, we contrapose the difficulty of feature extraction of small targets in high altitude remote sensing, increasing the detection scales from three to four. In addition, in order to avoid gradient fading, the 'Dense blocks' we proposed were used to replace the five convolutional layers in each detection layer. We can see from Tables 9–11 and Figure 10 that the MRFF-YOLO we proposed is superior to other state-of-the-art algorithms in remote sensing target detection. Since MRFF-YOLO was provided based on YOLO-V3, Tables 12–14 in ablation experiments showed that each module we proposed was valid for improving the accuracy of remote sensing target detection. A slight loss in detection speed is acceptable. The comparison of detection effect revealed that MRFF-YOLO performed better than YOLO-V3 in detecting densely distributed targets with a small size in remote sensing images. In general, our approach is more suitable for remote sensing target detection than YOLO-V3 and other classical target detection models, and it basically meets the requirement of real-time detection. In further work, other networks based on receptive field amplification will be researched.

**Author Contributions:** D.X. Methodology, software, provided the original ideal, finished the experiment and this paper, collected the dataset. Y.W. contributed the modifications and suggestions to the paper, writing—review and editing. All authors have read and agreed to the published version of the manuscript.

**Funding:** This research was funded by the National Nature Science Founding of China under Grant 61573183; Open Project Program of the National Laboratory of Pattern Recognition (NLPR) under Grant 201900029.

**Acknowledgments:** The authors wish to thank the editor and reviewers for their suggestions and thank Yiquan Wu for his guidance.

**Conflicts of Interest:** The authors declare no conflict of interest.

## Abbreviations

The abbreviations in this paper are as follows:

| | |
|---|---|
| YOLO | You Only Look Once |
| MRFF | Multi-Receptive Fields Fusion |
| CV | Computer Version |
| IOU | Intersection-Over-Union |
| FC | Full Connected Layer |
| FCN | Full Convolutional Network |
| CNN | Convolutional Neural Network |
| GT | Ground Truth |
| RPN | Region Proposal Network |
| FPN | Feature Pyramid Network |
| ResNet | Residual Network |
| DenseNet | Densely Connected Network |
| UAV | Unmanned Aerial Vehicle |
| SPP | Spatial Pyramid Pooling |
| NMS | Non-Maximum Suppression |
| TP | True Positive |
| FP | False Positive |
| FN | False Negative |
| TN | True Negative |
| AP | Average Precision |
| mAP | Mean Average Precision |
| FPS | Frames Per Second |

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
