# Peer review of "MRFF-YOLO: A Multi-Receptive Fields Fusion Network for Remote Sensing Target Detection"

_remotesensing, doi:10.3390/rs12193118_

Round 1
Reviewer 1 Report
The article entitled "MRFF-YOLO:A Multi-Receptive Fields Fusion Network for Remote Sensing Target Detection"
This article presents a very active project on which many groups are working. Here is presented a possible support to try to reduce some limitations of this approach. The article is presented well and completely.
Author Response
Thank you for your affirmation of the paper
Reviewer 2 Report
In this paper an innovative YOLO V3 based algorithm has been proposed. The paper includes a complete state of the art together with an adequate explanation of the contribution with conclusions supported by the results.
The main issue with the paper is the level of the use of the English language, which is unsatisfactory to be included in this publication. Authors whose primary language is not English are advised to seek help in the preparation of the paper.
Please also consider the following aspects in order to improve the paper:
- Titles of section 3 and 4 should be corrected. Section 3 is not referred to related work (related work is already included in the Introduction section) bu to the paper contribution itself. Section 4 title could be changed to "Results".
- Line 108 -> change "gird" to "grid".
- Lines 128-130 repeat lines 122-124.
- Define pi and ^pi in Equation 3.
- Line 205 -> Change "lay" to "layer".
- Lines 287-295 define sizes for anchor boxes of RSOD and UCS-AOD datasets. These sizes may be better included in a table.
- Lines 298-299: Define in these lines acronyms for true/false negative/positives, even if these are aincluded in Acronyms section.
- Tables 9, 10, 11, 12 and 13 include the header Metric (%) which seems to be referred to mAP (and when mAP appears as header seems to be an average of several frames). Please review this issue.
Author Response
1,I have modified the problem of language.
2,I have changed the titles of section3 and sectioni4.
3,I have changed 'gird' to 'grid' in line 105.
4, I have removed the repetitive parts.
5,I have defined them in line 147.
6,I have changed 'lay' to 'layer' in line 202.
7,I have added a table in line 295.
8, I have defined them in line 300-line 304.
9,I have changed 'Metric' to 'AP' in Tables 10-14
Reviewer 3 Report
Good paper however some minor points should be addressed:
- The paragraph talking about yolo in the introduction should be reduced (74-83)
- The performance reported should be compared with the state of art.
- I believe that more metrics could be used to access the performance FP and FN are not sufficient.
Author Response
1, I have reduced the introduction of YOLO in line 74-80.
2, The algorithms compared with our approach are all classical algorithms and their improved versions. They also include the latest proposed ones. So from my point of view, they are typical and state-of-the-art.
3, I have added more metrics.